Catch, bycatch and discards of the Galapagos Marine Reserve small-scale handline fishery

Zimmerhackel Johanna S. 1 j.zimmerhackel@yahoo.de
Schuhbauer Anna C. 1 2
Usseglio Paolo 3 4
Heel Lena C. 1 5 6
Salinas-de-León Pelayo 1
1 Department of Marine Science, Charles Darwin Research Station , Puerto Ayora, Galapagos Islands , Ecuador
2 Fisheries Center, Fisheries Economic Research Unit, The University of British Columbia , Vancouver, British Columbia , Canada
3 Fisheries Ecology Research Lab, University of Hawaii at Manoa , Hawaii , USA
4 Fundación In-Nova , Toledo, Castilla la Mancha , Spain
5 Institute of Ecology and Environmental Chemistry, Leuphana University Lüneburg , Lüneburg , Germany
6 Leibniz Center for Tropical Marine Ecology , Bremen , Germany
Johnson Magnus
Electronic publication date: 2015 Jun 9
Publication date: 2015
Volume: 3
Electronic Location ID: e995
Received 2015 Feb 11; Accepted 2015 May 12
Copyright: © 2015 Zimmerhackel et al.
Copyright year: 2015
Copyright holder: Zimmerhackel et al.
License: This is an open access article distributed under the terms of the Creative Commons Attribution License, which permits unrestricted use, distribution, reproduction and adaptation in any medium and for any purpose provided that it is properly attributed. For attribution, the original author(s), title, publication source (PeerJ) and either DOI or URL of the article must be cited.
License URL: https://creativecommons.org/licenses/by/4.0/

Keywords: Galapagos marine reserve, Small-scale fisheries, Bycatch, Multispecies fisheries, Bycatch mortality, Handline fishery, Galapagos sailfin grouper, Regulatory discards, Discards, Interview suveys

Funding: Galapagos Conservation Trust Lindblad National Geographic Fund LX 04-14 WildAid Helmsley Charitable Trust 2015PG-CON001 This study was funded by the Galapagos Conservation Trust, the Lindblad National Geographic Fund (LX 04-14), WildAid, and the Helmsley Charitable Trust (2015PG-CON001). The funders had no role in study design, data collection and analysis, decision to publish, or preparation of the manuscript.

==============================
Fisheries bycatch is a significant marine conservation issue as valuable fish are wasted and protected species harmed with potential negative ecological and socio-economic consequences. Even though there are indications that the small-scale handline fishery of the Galapagos Marine Reserve has a low selectivity, information on its bycatch has never been published. We used onboard monitoring and interview data to assess the bycatch of the Galapagos handline fishery by estimating the bycatch ratio, determining species compositions of landings and bycatch, identifying fishers’ reasons for discarding certain individuals, and revealing historical trends in the bycatch ratio. The estimated bycatch ratio as a function of biomass of 0.40 and a diverse species composition of target catch and bycatch confirmed the low selectivity of this fishery. Most individuals were not landed for economic motivations, either because species (77.4%) or sizes (17.7%) are unmarketable or for regulatory reasons (5.9%). We found that bycatch contributes to growth overfishing of some target species because they are discarded or used as bait before reaching their first maturity. Moreover, over half of interviewees perceived a historical decrease in bycatch ratios that was explained by a diversification of the target catch due to the reduction in abundance of the traditionally most important target species. As some target species show signs of overfishing and to date there are no specific regulations for the finfish fishery species in place, we recommend the implementation of a series of management measures to protect critical life stages of overexploited species and to improve the selectivity of the Galapagos handline fishery.

Introduction

The role of bycatch in global fisheries has become a significant marine conservation issue, especially in areas where serious ecosystem degradation has already been observed (Harrington, Myers & Rosenberg, 2005). Bycatch is commonly referred to as the incidental catch of non-target species and is divided into the portion of the catch that is discarded because species or sizes are not marketable or of lower economic value (economic discards), and catch that is discarded due to regulatory restrictions (regulatory discards) e.g., protected species or certain sizes (Dunn, Boustany & Halpin, 2011; National Marine Fisheries Service, 2011). Bycatch, when discarded, causes significant waste of natural resources and is of particular concern when the populations of the captured species are already severely overfished or threatened (Alverson et al., 1994; National Marine Fisheries Service, 2011). However, discards was proven to also have positive effects on the marine fauna that feeds on discards including lobsters, marine mammals, birds and sharks (Blaber & Wassenberg, 1989; Wassenberg & Hill, 1990; Saila, Nixon & Oviatt, 2002; Bozzano & Sardà, 2002; Grabowski et al., 2010). Bycatch has serious ecological consequences not just for the species caught, but also for entire marine ecosystems (Dayton et al., 1995; Crowder & Murawski, 1998; Saila, Nixon & Oviatt, 2002; Dulvy, Sadovy & Reynolds, 2003; Kappel, 2005). Ecological impacts on community structure and fishery productivity are the result of increased fishing mortality of species that are important to shape the ecosystems such as species at high trophic levels (Myers et al., 2007; Shester & Micheli, 2011) which can cause alterations in species assemblages and widespread community impacts via trophic cascades (Pauly et al., 1998; Lewison et al., 2004). In marine fisheries, bycatch implications include the negative economic impacts of foregone income due to discards of undersized individuals of commercially valuable species. Furthermore, the costs associated with discarding non-commercial species (Pascoe, 1997; Bjorkland, 2011; Dunn, Boustany & Halpin, 2011), also includes the creation of a negative public image of fishers for wasting resources and for bycatching certain charismatic animals such as dolphins or marine turtles (Hall, 1999). Because of the high impact of bycatch in fisheries, Bjorkland (2011) stated that “the ecological, economic and social costs of bycatch in fishing activities are increasingly indefensible to governments, fishing interests, marine scientists and ocean activists”, making it necessary to establish appropriate measures and finding alternative gear to successfully reduce the impact of bycatch on a global scale.

Bycatch in small-scale fisheries

Most bycatch studies have focused on industrial fisheries, leaving a lack of information regarding small-scale fisheries, in particular towards effort, catch and bycatch (Lewison et al., 2004; Moore et al., 2010). Small-scale fisheries are often described to be more selective and potentially more sustainable than industrial fisheries and to be therefore the most sustainable option for the utilisation of coastal marine resources (Chuenpagdee et al., 2006; Jacquet & Pauly, 2008). However, recent studies show that bycatch in small-scale fisheries can have severe ecological impacts, and if scaled to per-unit of total catch they can be comparable to industrial fisheries (Soykan et al., 2008; Shester & Micheli, 2011; Parker & Tyedmers, 2014). Moreover, small-scale fisheries are generally understudied and often unregulated (Mora et al., 2009; Davies et al., 2009; Chuenpagdee, 2011). As small-scale fisheries encompass 44% of the world’s 50 million fishers and provide over half of the total global fisheries production (Berkes et al., 2001; Chuenpagdee et al., 2006; Teh & Sumaila, 2013), this knowledge gap represents a major challenge to sustainable fisheries management and the conservation of threatened species, especially in tropical fisheries of developing countries (Moore et al., 2010).

The Galapagos handline fishery

The Galapagos Archipelago did not have a consistent human presence until the 1930s (Reck, 1983; Danulat & Edgar, 2002; Castrejón, 2011). Since then, the highly productive and diverse marine ecosystems of the archipelago have been increasingly threatened by human activities, reflected by the exponential increase in the human population from 6,119 inhabitants in 1962 to 25,000 in 2010 (INEC, 2011), along with an increase in the number of tourists, which reached over 200,000 visitors per year in 2013 (DPNG, 2014). To ensure the sustainable economic development and protect the biodiversity of Galapagos, the 133,000 km2 Galapagos Marine Reserve (GMR) was established in 1998. While industrial fishing was banned within the reserve, fishing rights were granted exclusively to the local small-scale fishing sector. The GMR was divided into functional zones including no take zones, where fishing activities are prohibited. These zones make up 18% of the coastline (Heylings, Bensted-Smith & Altamirano, 2002). Additionally, the implemented Organic Law for the Special Regimen for the Conservation and Sustainable Development of Galapagos (LOREG) includes regulations for iconic species such as sharks, marine mammals and sea horses, which are excluded from extractive activities, and if caught unintentionally, have to be returned to their natural environment. However, there is evidence that the established artisanal fishery caused major impacts upon fishing resources (Burbano et al., 2014; Schiller et al., 2014). The collapse of the sea cucumber fishery in the early 2000s represents the most severe example (Hearn, 2008; Wolff, Schuhbauer & Castrejón, 2012). The multispecies handline fishery (locally called “empate”) is traditionally the most important in Galapagos. Until the 1960s, fishers had no access to refrigeration and therefore preserved fish by salting and drying it. Fish were then exported to the mainland where they formed the main ingredient of “fanesca”, a traditional Ecuadorian dish served at Easter. While presently the handline fishery for fresh demersal finfish occurs all year round to supply local markets, the main market still remains the exported salt-dried finfish to serve the ongoing demand for “fanesca”, and is caught during the hot season (December to April). The selectivity of the handline fishing method has been described as both low for species and size ranges (Nicolaides et al., 2002; Peñaherrera & Hearn, 2008), but conversely also as fairly selective (Ruttenberg, 2001). However, to date no information on bycatch for this fishery has been published. Studies have demonstrated that the handline fishery has caused an impact on several exploited fish stocks, and revealed a dramatic shift in the volume of fish landings and in the species composition of the handline fishery (Ruttenberg, 2001; Burbano et al., 2014; Schiller et al., 2014). Despite the increasing evidence that there is a continuous trend of overexploitation of target species, to date there has been no particular management plan in place for any of these species. As the fishing sector sustains fishers’ livelihoods and plays a significant role in the regional culture it is crucial to prevent a further decline in the key target species, such as the regional endemic sailfin grouper Mycteroperca olfax which is considered vulnerable on the IUCN red list of threatened species (Castrejón, 2011). A better understanding of the complete catch of this fishery, including bycatch species and their sizes, is therefore an important step towards a sustainable Galapagos handline fishery.

Aims of this study

The aims of this study are to quantify the catch selectivity and bycatch ratios of the Galapagos handline fishery. This information will help to establish a knowledge baseline from which changes in bycatch ratios can be monitored and to inform decision making processes for future fisheries management plans. We then analyse the social component of this multispecies fishery by identifying the fishers’ reasons for discarding certain individuals. Moreover, we hypothesize that changes in the availability of key target species have resulted in changes in the fishers’ decision making process of whether to keep or discard a specimen. In order to test this hypothesis, we use interview surveys to evaluate historical trends in the bycatch ratio and reasons for potential changes in bycatch levels.

Materials and Methods

Fishery observations

We monitored artisanal handline fishing trips with onboard observers from February to May 2012. Fishers were asked for permission to take us on their fishing trips and a total of 62 fishers (15.5% of all Galapagos active fishers) in 14 fishing boats agreed to let us on board. The handline technique consists of a monofilament line weighted with lead and several short extensions of propylene line each with one hook (Danulat & Edgar, 2002). Fishing depths ranged from 15 to 200 m, with trip durations lasting from one to two days and an average duration of 8 h (SD = 6.5). Departure and arrival date and time, vessel horsepower and number of fishers on board were recorded for each trip. During each fishing trip, fishers actively looked for promising bottom structure and fished for several minutes on selected sites before moving to the next. We recorded the effective fishing time at each of these sites as the interval starting when the first line was cast and ending when the last line was out of the water. Start and stop time, geographical position, number of hooks and lines in the water, number of fishers, water depth, bait and capture time were recorded at each site. The study area with all monitored fishing sites is shown in Fig. 1. Total lengths of all individuals were recorded and converted to weight using available length-weight relationships (Froese & Pauly, 2000). If no length-weight relationship was available, these were obtained by means of regression analysis on our catch data as suggested by Lima-Junior, Cardone & Goitein (2002). Whenever a species could not be identified by observers and fishers, we took a picture of the individual and identified the species on land with the help of local fishery experts and Fishbase.org (Froese & Pauly, 2000). Catch was categorized according to the bycatch definition of the US National Marine Fisheries Service (MSA 1996), such that all individuals that are either sold or used for personal consumption are categorized as landings, while all other individuals are bycatch. We furthermore distinguished different bycatch categories between bycatch survival (individuals that were discarded alive) and bycatch mortality (individuals that were discarded dead or used as bait). Additionally, the condition of individuals when released was recorded and their release observed. Whenever possible, the post-release mortality was noted, but could not be measured consistently for all discarded individuals.

Figure 1 Geographic position of the study site in the Galapagos Marine Reserve.

Geographic position of the study site with the fishing ports (stars) and the monitored fishing sites (dots).

Bycatch estimates

Landings and bycatch were expressed as numbers of individuals and biomass (kg). Additionally, for each of the defined landing and bycatch categories, biomass percentages were calculated. The bycatch ratio (BCR) is defined as the ratio of bycatch to total catch, whereby total catch equals landings plus bycatch. BCR was obtained as a function of abundance (BCRN) and biomass (BCRW).

Species composition

Species composition is shown as numbers of species categorized as landings or bycatch. We identified three reasons for fishers not landing an individual, and divided the bycatch accordingly into the three subcategories: species that are not lucrative because they have low or no market value were defined as “not marketable species”, small sized and therefore not lucrative individuals of otherwise marketable species were defined as “not marketable sizes”, and bycatch of protected species was defined as “regulatory discard”. We report the average Total Length (TL) of each species represented in these categories as well as the bycatch ratio of each species (BCRS), defined as the ratio in which the number of individuals of each species belong to the bycatch.

Prediction of bycatch sizes

For exploited species for which an adequate sample size was obtained (n ≥ 100), a logistic regression model was used to estimate the probability of a fish being landed based on its size. Fish TL was summarized into 5 cm length categories. Proportion of fish considered as landed was calculated for each length category. The model followed the formula: Logitp=p1−p

where 1 − p is the probability that a given fish would not be landed. Confidence intervals of the parameters of the regression were estimated via bootstrapping with 100 iterations. Analyses were done using the R package FSA (Ogle, 2013). The resulting predictive model was used to estimate the size below which a fish would have an 80% probability of becoming bycatch (b80). We furthermore obtained the odds ratio of the model, which is the factor by which the probability of an individual to be landed increases with each 5 cm in TL. The b80 value was compared to the mean length at which species reach first maturity (Lm). Lm was estimated from the maximum length (Lmax) of the species using the following formula as suggested by Froese & Binohlan (2000). Log10Lm=−0.2713+Log10Lmax * 1.0260

Lmax was obtained from Fishbase for C. princeps (102 cm). For P. albomaculatus (65 cm) and P. clemensi (61 cm) we used the Lmax of our own data set because it was higher than the published Lmax from Fishbase.

Interview surveys

To obtain additional information about bycatch species and historical changes in bycatch composition and quantities, a total of 100 semi-structured interviews with fishers from Santa Cruz (26%) and from San Cristobal (74%) Islands were conducted representing approximately 25% of the 400 active fishers in the GMR. Because of the close relationship the fishers have with their environment, we used their experience and knowledge, as this information can fill important knowledge gaps including the abundance of fish stocks and perceived historical changes in the fishery (Johannes, Freeman & Hamilton, 2000; Murray, Neis & Johnsen, 2006; McCluskey & Lewison, 2008). From April to May 2012, we approached fishers from Santa Cruz and San Cristobal Islands and asked them for permission to carry out in-person interviews. Because interviewers had already worked closely with fishers and guaranteed their anonymous status, it was possible to gain the fishers’ trust. Therefore no fishers rejected the participation and answers are believed to be reliable. To avoid any influence on fishers’ responses, interviews were carried out with one fisher at a time. Interviewed fishers were asked to suggest fellow fishers who could be interviewed, who we then approached at the fishing dock in order to ask for their participation in the interview. Our use of this snowball sampling technique (Goodman, 1961) helped ensure that an adequate number of interviews (n ≥ 78, N = 400, confidence level = 95% and margin of error = 10%) were completed. In order to identify species that are commonly caught as bycatch, we asked fishers: “Can you name species that you discard or use as bait while fishing with handlines?” We furthermore asked: “For what reason do you not land these species?”. A Pearson’s chi square test was used to test for interactions among the answers given and the island of residence of the fishers.

Additionally, we asked fishers about their perceptions of historical changes with the questions: “Do you perceive any changes in the amount of individuals that you either discard or use as bait during your working life?”, “Did the bycatch ratio decrease, increase or stayed the same?” and “Please give an explanation for your perception.” We used an open interview as it has been proven to provide a much more detailed description of the answers provided (Jackson & Trochim, 2002). Answers about most common bycatch species, reasons for not landing these species, historical changes in bycatch and reasons for changes given by fishers were manually coded, each code representing one explanation that fishers gave. We chose this approach because answers to open questions can vary in the description and human analysers are able to interpret the subtleties in answers to categorize and code them. We then calculated the percentages of each coded answer.

The research was approved by the Galapagos National Park under the annual research plan of the Charles Darwin Foundation (POA 2012, number 86).

Results

Bycatch estimates

A total of 22 fishing trips were conducted, resulting in 153 h at sea and 94 h of effective fishing time. During fishing trips, 297 sites were visited and 1,279 fish with a total combined biomass of 2.1 metric tonnes. Fractions of landing and bycatch categories are shown as a function of biomass in Fig. 2. Total bycatch weighted 883 kg (n = 543), resulting in a BCRN of 0.43 and a BCRW of 0.40.

Figure 2 Catch categories and their fractions of the total catch biomass.

Fraction of the total biomass for landings and bycatch (outer circle) and the fractions of the according subcategories (inner circle) for landings (dashed green): personal use (dark green) and sold (light green), and for bycatch (dashed blue): bycatch mortality (dark blue) and dis carded alive (light blue). The bycatch mortality is divided by the fraction discarded dead and the fraction used as bait.

Landing composition

We observed a total of 36 species caught by the Galapagos handline fishery. Landings were composed of 17 fish species belonging to seven families. Of these, five species were landed exclusively and the remaining 12 species were sometimes landed and sometimes discarded or used as bait. Landings were dominated by fish of the family Serranidae, which was represented by eight species and made up for 68% of the landed biomass. The Galapagos sailfin grouper (M. olfax) and the camotillo (Paralabrax albomaculatus) were the most landed species constituting 40% and 13% of all landed biomass, respectively. Other common target species were the ocean whitefish (Caulolatilus princeps) and the mottled scorpionfish (Pontinus clemensi) representing 13% and 10% of the landed biomass, respectively. While the first two species are fished in depths ranging from 15 to 40 m, the latter two species are targeted in deeper waters of up to 200 m. Fishers used 7% of landed biomass for their personal purposes which were represented by the five species (from highest to lowest occurrence) C. princeps, M. olfax, P. clemensi, P. albomaculatus and the starry grouper Epinephelus labriformis. Descriptive statistics of catch including the number of individuals per species, average size and bycatch ratios are shown in Table 1.

Table 1 Marketable species that were landed during onboard monitoring.

Shows the numbers of individuals landed (N), their average total length with its standard deviations (Av. TL ± SD) and the bycatch ratio of each particular species (BCRs).

Family	Scientific name	Common name	N	Av. TL ± SD (cm)	BCRs	
Serranidae	Mycteroperca olfax *	Galapagos sailfin grouper	368	45.9 ± 8.5	0	
Serranidae	Cratinus agassizi	Grazery threadfin seabass	16	59.8 ± 11.5	0	
Serranidae	Epinephelus mystacinus	Misty Grouper	2	83.0 ± 5.0	0	
Carangidae	Caranx caballus	Green jack	1	49.0 ± 0.0	0	
Lutjanidae	Hoplopagrus guentheri	Barred Snapper	1	72.0 ± 0.0	0	
Labridae	Semicossyphus darwini	Galapagos sheephead wrasse	37	51.4 ± 7.4	0.08	
Malacanthidae	Caulolatilus princeps	Ocean whitefish	88	42.5 ± 5.0	0.21	
Serranidae	Paralabrax albomaculatus *	Camotillo	85	44.9 ± 7.5	0.24	
Scorpaenidae	Pontinus clemensi *	Mottled scorpionfish	106	45.3 ± 7.4	0.25	
Sparidae	Calamus taurinus *	Galapagos porgy	6	38 ± 4.1	0.25	
Malacanthidae	Caulolatilus affinis	Bighead tilefish	2	48.5 ± 3.5	0.33	
Serranidae	Hemilutjanus macrophthalmos	Grape eye seabass	3	58.3 ± 1.3	0.4	
Carangidae	Caranx sexfasciatus	Bigeye trevally	1	46.0 ± 0.0	0.5	
Serranidae	Epinephelus cifuentesi	Olive grouper	2	64.5 ± 21.5	0.6	
Serranidae	Epinephelus labriformis	Starry grouper	6	38.7 ± 3.0	0.89	
Haemulidae	Anisotremus scuderii	Peruvian grunt	6	31.3 ± 3.1	0.93	
Haemulidae	Anisotremus interruptus	Burrito grunt	3	32.3 ± 2.1	0.98	
Notes.

* Denote endemic species to Galapagos.

Bycatch composition and sizes

We found 31 species that were caught unintentionally out of which 19 species were always discarded or used as bait. Out of the 43% of bycatch (number), we could distinguish between three different reasons for fishers not landing certain individuals: regulatory discards and individuals not marketable due to the species or their sizes. Regulatory discard included 26 juvenile sharks (23 Carcharhinus galapagensis and 3 Triaenodon obesus) as well as two sea lions (Zalophus wollebaeki). Protected species made up for 5.9% of all caught individuals as bycatch. Eighteen species were not landed because they were considered not marketable species constituting 77.4% of all caught individuals as bycatch. The most frequently caught unmarketable species were: the burrito grunt Anisotremus interruptus, the peruvian grunt Anisotremus scuderii, E. labriformis and the greybar grunt Haemulon sexfasciatum. Twelve species representing the remaining 17.7% of the bycatch were not landed because fishers considered the size of individuals too small to be economically valuable. The number of individuals per species caught as bycatch, average sizes and bycatch ratio are shown in Table 2.

Table 2 Not marketable species, not marketable sizes and regulatory discards that were recorded during onboard monitoring.

Shows numbers of specimens (N), their average total length with its standard deviations (Av. TL ± SD) and the bycatch ratio of each particular species (BCRs).

Family	Scientific name	Common name	N	Av. TL (cm) ± SD	BCRs	
Not marketable species	
Haemulidae	Haemulon sexfasciatum	Greybar grunt	29	30.0 ± 4.9	1	
Lutjanidae	Lutjanus viridis	Blue and gold snapper	19	26.2 ± 3.6	1	
Serranidae	Paranthias colonus	Pacific creolefish	17	30.6 ± 4.3	1	
Sphyraenidae	Sphyraena idiastes	Pelican barracuda	11	59.3 ± 9.9	1	
Haemulidae	Haemulon scudderi *	Grey grunt	6	32.0 ± 3.5	1	
Balistidae	Balistes polylepis	Finescale triggerfish	5	45.6 ± 1.0	1	
Balistidae	Sufflamen verres	Orangeside triggerfish	5	37.4 ± 5.2	1	
Scorpaenidae	Scorpaena mystes	Pacific spotted scorpionfish	2	28.0 ± 0.0	1	
Synodontidae	Synodus lacertinus	Banded lizardfish	2	34.0 ± 7.0	1	
Kyphosidae	Girella freminvilli	Dusky chub	1	35.0 ± 0.0	1	
Muraenidae	Murraena sp.	Moray eel	1	60.0 ± 0.0	1	
Scombridae	Scomberomorus sierra	Pacific Sierra	1	90.0 ± 0.0	1	
Scorpaenidae	Scorpaena histrio	Bandfin scorpionfish	1	33.0 ± 0.0	1	
Serranidae	Serranus psittacus	Barred serrano	1	13.0 ± 0.0	1	
Tetradontidae	Sphoeroides annulatus	Bullseye puffer	1	27.0 ± 0.0	1	
Haemulidae	Anisotremus interruptus *	Burrito grunt	191	33.2 ± 5.0	0.98	
Haemulidae	Anisotremus scuderii	Peruvian grunt	81	32.2 ± 2.9	0.93	
Malacanthidae	Caulolatilus affinis	Bighead tilefish	3	45.7 ± 4.5	0.33	
Not marketable size	
Serranidae	Dermatolepis dermatolepis	Leather bass	1	46.0 ± 0.0	1	
Serranidae	Epinephelus labriformis	Starry grouper	51	36.2 ± 3.8	0.89	
Serranidae	Epinephelus cifuentesi	Olive grouper	3	35.0 ± 4.1	0.6	
Carangidae	Caranx sexfasciatus	Bigeye trevally	1	43.0 ± 0.0	0.5	
Serranidae	Hemilutjanus macrophthalmos	Grape eye seabass	2	49.0 ± 1.0	0.4	
Scorpaenidae	Pontinus clemensi *	Mottled scorpionfish	35	31.2 ± 5.3	0.25	
Sparidae	Calamus taurinus *	Galapagos porgy	2	36.5 ± 6.5	0.25	
Serranidae	Paralabrax albomaculatus *	Camotillo	27	36.2 ± 6.1	0.24	
Malacanthidae	Caulolatilus princeps	Ocean whitefish	24	38.3 ± 6.1	0.21	
Labridae	Semicossyphus darwini	Galapagos sheephead wrasse	3	43.0 ± 5.0	0.08	
Regulatory discard	
Carcharhinidae	Carcharhinus galapagensis	Galapagos shark	23	74.4 ± 8.4	1	
Carcharhinidae	Triaenodon obesus	Whitetip reef shark	3	110.0 ± 0.0	1	
Otariidae	Zalophus wollebaeki	Californian sea lion	2	n.a.	1	
Notes.

* Denote endemic species to Galapagos.

The species P. albomaculatus, C. princeps and P. Clemensi were not only some of the most important target species in landings, they also were some of the most frequently caught bycatch species. Those three species made up five, four and two percent of all bycatch biomass, respectively. The biomass of C. princeps was mostly landed (79%), but partly used as bait (19.6%), partly discarded dead (1.2%) and to a small extent discarded alive (0.6%). Of the total biomass of P. albomaculatus, 76% was landed, 16.1% was used as bait, 8.0% discarded dead and only 0.9% was discarded alive. Finally, 75% of the caught biomass of P. clemensi was landed, 22.7% used as bait and 2.1% was discarded. No individuals of this species were discarded alive. An adequate sample size (n ≥ 100) for these three species allowed us to apply a logistic regression model which predicted the size below which individuals have a 80% chance to become bycatch. Results of this model are indicated in Fig. 4.

Interview surveys

The interviewed fishers’ ages ranged from 19 to 80 years, with an average of 43.0 years (SD = 11.9). While 42% of interviewed fishers were born in the Galapagos Islands, the remaining 58% were originally from mainland Ecuador. Of the 43 different species caught as bycatch, the reasons given for not landing 27 of these species was that they were not marketable species, whereas the other 14 were considered as bycatch when caught under a certain size to be marketable. Additionally, five of these species were discarded for both these reasons. Haemulidae (79%) and Serranidae (37%) were the most frequently mentioned families, represented by six and nine different species, respectively.  The most common bycatch species mentioned by fishers were A. interruptus (39%), A. scuderii (26%), E. labriformis (24%) and Sphoeroides annulatus (21%). Furthermore, 73% of fishers stated that they occasionally bycaught protected species. Of these, 68% identified sharks as bycatch with 29% of these were identified as C. galapagensis, 2% as Carcharhinus falciformes, 1% as T. obesus, while the remaining 36% did not specify the species. Rays were mentioned by 20% of fishers, turtles by 14%, sea lions by 13% and marine birds by 3% (Fig. 3). There was no significant difference between the number of species reported by fishers from the two different islands of residence based on the Pearson’s chi square test (p = 0.45).

Figure 3 Logistic regression model results showing the probability of an individual to belong to bycatch or to landings depending on the individuals’ total length.

Shows the probability of an individual to belong to bycatch (0) or to landings (1) depending on the individuals’ total length. The dashed blue line indicates the b80, the dashed grey line indicates the mean size of first maturity (Lm) of the species: (A) C. princeps (n = 112, b80 = 42.7 cm TL, Lm = 61.6 cm TL, odds ratio = 1.16); (B) P. albomaculatus (n = 112, b80 = 39.2 cm TL, Lm = 36.3 cm TL, odds ratio = 1.24); (C) P. clemensi (n = 141, b80 = 38.2 cm TL, Lm = 38.8 cm TL, odds ratio = 3.25e–7).

Figure 4 Percentage of responses of interviewees (n = 100) for each mentioned taxa as well as the reasons of fishers to not land these taxa.

Reasons to not land taxa are unmarketable species (dark blue), unmarketable size (light blue) and regulatory discard (green).

Perception of historical changes of bycatch

Results from interviews revealed that 52% of fishers perceived a decrease in bycatch throughout their working life mostly attributed to general decreases of fish abundance (44%), shift in species composition of landings (21%) or a change in their main fishing gear (13%). On the other hand, eight percent of interviewees stated that they observed an increased amount of discards, which they explained with changes in fishing regulations. A third (31%) of fishers stated that there was no change and 9% did not answer this question.

Discussion

This study provides the first insight into the selectivity of the Galapagos handline fishery. Our results suggest that Galapagos small-scale fisheries are not necessarily more selective than industrial fisheries as has been found elsewhere (e.g., Shester & Micheli, 2011). We found the bycatch of the handline fishery to consist of a fairly diverse fish fauna where most specimens are discarded due to economic motivation, and to a lesser extent because of regulatory restrictions. Undersized individuals of some commercially exploited species suffer bycatch mortality contributing most probably to their overexploitation. Moreover, interviews revealed that the overexploitation of the commercial species caused a diversification of the catch composition which resulted in a historical change in the bycatch level towards lower bycatch ratios.

Species composition

The diverse catch composition of landed fish confirmed a low selectivity of this fishery and revealed that fishers consider a large part of their catch as target species. However, monitoring and previous studies on this fishery focused mainly on the Galapagos sailfin grouper (Schiller et al., 2014). Given the lack of attention on other exploited species and missing management measures for any fish species in the GMR, most of the species caught are scarcely measured and poorly documented. A management plan for these species is urgently needed and should take into consideration the multispecies character of this fishery rather than focusing on single species management.

The overall bycatch of protected species recorded in this study was considerably low. However, results can be biased towards lower bycatch ratios and mortality caused by the observer effect, which occurs when fishers tend to follow a best practice fishing attitude during onboard monitoring, as opposed to un-observed fishers (Hall, 1999). Our results from both onboard observations and interview surveys confirm speculations that sharks are occasionally caught and discarded by the Galapagos handline fishery (Jacquet et al., 2008; Castrejón, 2011). Sea lions scavenging around fishing gear increase their own susceptibility to incidental capture. The two by-caught sea lions got hooked on the fishing gear, while trying to feed on the captured fish and got injured because fishers hit them with a wooden plank with a nail attached to expel them. Even though this study did not detect any mortality of sharks and sea lions, there are indications that bycatch mortality of protected species occurs as sea lions are occasionally found dead, showing evidence of having died due to unnatural causes (Denkinger, Quiroga & Murillo, 2014). Fishers see sharks and sea lions as competitors for marine resources and therefore as a threat to their livelihood (FT, LA, JG, FV, CC, WB, pers. comm., 2012). Previous studies point out that discards of protected species might be under-reported, because fishers fear negative consequences when accurately reporting bycatch of these taxa (National Marine Fisheries Service, 2004; Lewison et al., 2004). However, the high number of interviewed fishers who stated that they catch protected taxa by accident suggests that fishers answered our questions accurately.

Bycatch estimates

The estimated bycatch ratios of 0.40 (biomass) and 0.42 (numbers) are comparable to current global fisheries bycatch estimates of 40.4% (Davies et al., 2009). A study in Baja California, Mexico found strong varying discard rates for different artisanal fishing gears (0.11% for fish traps, 15.1% for lobster traps, 18.5% for drift gillnet and 34.4% for set gillnets) (Shester & Micheli, 2011). Even though the results of these studies are due to the assessments of different fishing techniques and species, and therefore not directly comparable with our results, it is interesting to note that the bycatch ratio of the Galapagos handline fishery is similar or higher than the ratios of the other fisheries studied.

Species that suffered bycatch mortality consisted mostly of grunts and small sized individuals of economically valuable species. Bycatch of non marketable undersized individuals represents not only a waste of resources because specimens are being harvested before reaching their maximum yield per recruit, but it also contributes to growth overfishing of the most exploited species (Alverson et al., 1994). P. albomaculatus reaches first maturity at 36.3 cm TL which lays below the b80 of 39.2 cm indicating that there is no market for undersized individuals. However, immature individuals still suffer bycatch mortality because they are being used as bait which is of special concern, because P. albomaculatus is endemic to the Galapagos and classified as endangered on the IUCN red list of threatened species (Robertson et al., 2010). While the Lm and the b80 of P. clemensi is almost equal also indicating that fishers do not land immature individuals, the Lm of C. princeps was much higher than the b80. This is concerning because individuals are being landed before being able to reproduce which increases the chance for the population to suffer growth overfishing. The lack of knowledge about the biology of these species impedes a proper risk assessment, which is necessary for their effective management.

Individuals that are discarded alive are still vulnerable as the interaction with the fishing gear can negatively affect the survival of the fish and lead to post-release mortality (Ryer, Ottmar & Sturm, 2004). Among the reasons for this mortality are decompression sickness, deficits in swimming ability, feeding, and a higher vulnerability to predators (Davis, 2002). As delayed mortality was impossible to observe from onboard the fishing boat, the bycatch mortality might be higher than estimated here.

Historical changes of bycatch

Our results about historical changes of bycatch levels support signs of negative impacts on exploited species imposed by this handline fishery, which already go back to the 1980s (Reck, 1983; Nicolaides et al., 2002; Burbano et al., 2014; Schiller et al., 2014). The consequences are characterized by an alteration of the species assemblages in form of a strong decline in abundance and average size of apex-level fish, such as the targeted groupers (Reck, 1983; Bustamante, 1998; Nicolaides et al., 2002; Edgar et al., 2010; Schiller et al., 2014), which drives fishers to target more species and smaller sized fishes. Besides, consequences of the decline of top predators also affects marine communities as sites with high fishing pressure show a lower variability in the fish community structure indicating significant changes in the functioning of coastal marine environments of the archipelago (Ruttenberg, 2001). Diversification of fishing gear and an increasing demand for fresh fish for local consumption are also reasons for the diversification of target species and the decreasing fraction of groupers caught with handlines within the finfish fishery of Galapagos (Castrejón, 2011). This is supported by seven percent of fishers who stated that their bycatch ratio decreased because they changed their fishing gear. Species like mullets (e.g., Xenomugil thoburni and Mugil galapagensis) caught with beach seine nets and pelagic species (e.g., Thunnus albacares and Acanthocybium solandri) caught trolling that were only occasionally caught in the late 1970s now make up 58% of total landings (Schiller et al., 2014).

The bycatch estimates, biomasses and catch compositions obtained by this study might be season-specific or even variable over the years. Therefore, results may only be representative for the observed time in this study months (from February to May 2012). Additionally, not all of the Galapagos archipelago was monitored so there might be regional differences that are not considered in the current investigation. Hence, further investigations on a larger spatial scale and over a longer time period are recommended.

Management suggestions

As multispecies fisheries target many different species, the general goal of increasing the selectivity of a fishery may not always be appropriate. Instead, the focus may rather be on reducing the bycatch of overexploited, threatened and protected species (Gillett, 2011). Furthermore, negative effects such as post release mortality on threatened bycatch species should be minimized and measures should involve appropriate implementation costs and should not affect fishing operations negatively (Sales et al., 2010). Here, we suggest management regulations towards a more sustainable Galapagos multispecies handline fishery.

Unravelling the problem of fisheries’ selectivity is often associated with the improvement of gear settings (Broadhurst, 2000; Bache, 2003). For example, the use of certain bait species was found to influence the bycatch of cod in the Northwest Atlantic haddock fishery (Ford, Rudolph & Fuller, 2008). Fishers from the Galapagos handline fishery stated that bait species are not equally selective and that the use of yellowfin tuna (Thunnus albacares) as bait seems to be related to the bycatch of sharks (JS Zimmerhackel, 2012, unpublished data). Also larger hook sizes have been proven to be more effective in capturing larger size classes of targeted fish (Ralston, 1990) and post-release mortality of groupers were found to be significantly lower when using circle hooks instead of J-hooks (Burns & Kerr, 2008). We therefore recommend experimental investigations into distinctive hook types, hook sizes and bait species in order to determine a gear setting that reduces the catch of unwanted species, sizes and post-release mortality, without negatively affecting the target catch.

Unfortunately, the lack of specific biological knowledge about the most exploited species of this fishery impedes a proper assessment of their population status. Therefore, critical life stages and spawning grounds of the main target species C. princeps, M. olfax, P. albomaculatus and P. clemensi should be assessed. Spatiotemporal closures of spawning aggregations should be taken into consideration in future zonification and management plans because the protection of critical life stages can effectively reduce the impact on threatened species (Beets & Friedlander, 1999; Lester & Halpern, 2008; Afonso, Fontes & Santos, 2011). The question of whether reducing or encouraging discards results in a more effective management of fisheries resources, is still under debate because regulations such as harvest restrictions, meant to protect target species, can raise the volume of discards (Diamond & Beukers-Stewart, 2011). However, the ultimate impact of bycatch on populations are influenced by the bycatch mortality (Davis, 2002). As groupers were proven to have a high post-release survivorship (Burns & Kerr, 2008) we suggest the implementation of minimum and maximum catch sizes and the exclusion of undersized individuals from the usage as bait to reduce their fishing mortality before reaching first maturity. Illegal fishing activities in regulated fisheries (such as the lobster and sea cucumber fishery) in the history of the GMR show that management regulations are often not respected by fishers (Hearn, 2008). The effectiveness of the marine reserve should therefore be studied. Furthermore, the suggested measures should be accompanied by plans to raise fishers’ awareness about bycatch related concerns and their implications for the sustainability of fish stocks.

Contrary to a common concern raised by the fishers, the reduction of fishing pressure on threatened target species does not necessarily have to be accompanied by a reduction of income. For example, integrating more resilient, faster growing non-target species in landings has been successfully adopted in a number of fisheries worldwide (Lobo et al., 2010; Rodríguez-Preciado, Madrid-Vera & Meraz-Sánchez, 2012). In the Mexican Pacific, bycatch species of the family Haemulidae such as Pomadasys panamensis have become an important part of the commercial catch from the fisheries (Rodríguez-Preciado, Madrid-Vera & Meraz-Sánchez, 2012). The fact that many species which presently are commonly consumed in the Galapagos handline fishery have often been discarded during previous decades indicates a certain flexibility and ability by the fishing sector and the consumer community to adapt to changes in their environment. This demonstrates that there is hope that new target species such as grunts (which together made up 51.1% of the bycatch biomass) could be accepted by both the fishers and consumers. However, the integration of new target species should ideally be accompanied by stock assessments on harvested species to prevent overfishing and all potential management alternatives should be evaluated on an ecological and socio-economic basis by including the main stakeholders and fishers in the solution finding process (Usseglio, Schuhbauer & Friedlander, 2014).

Conclusions

This information about bycatch of the Galapagos handline fishery revealed that this fishery targets a fairly high number of species and is not selective for species or size classes. Most individuals are not landed due to economic motivations, either because the species or the fish sizes are not marketable. Regulatory discards were observed to a lesser extent, indicating that protected species are not discarded very frequently. However, more than two thirds of interviewed fishers mentioned that they discard sharks. A more concerning result was the high number of small sized individuals of some target species, which mostly suffer bycatch mortality mainly because they are used as bait, which increases their overall fishing mortality. Moreover, interviews revealed a historical change in the bycatch level towards lower bycatch ratios that was explained by a diversification of the catch composition due to the overexploitation of some commercial species. As it becomes more evident that the most exploited target species of this fishery are overfished (Burbano et al., 2014; Schiller et al., 2014) and to date there are no regulations for any target species in place, our results demonstrate the need to integrate management measures in future management plans in order to minimize the fishing pressure on threatened and protected species.

Supplemental Information

Annex S1 Fitting parameters of the length-weight relationship for species where these information were not available in the literature

Fitting parameters (a and b) of the length-weight relationship and the number of individuals measured (n).

Click here for additional data file.

Annex S2 List of bycatch species that were mentioned by interviewed fishers once

Shows the species that were mentioned once by interviewees as well as the reasons to not land these species.

Click here for additional data file.

Supplemental Information 3 Raw data of onboard observations about catch and bycatch of the Galapagos handline fishery 2012

The data show the fishing sites (numbered), the geographic position of the fishing site (latitude and longitude), the date of the fishing trip, the time when the fishers started their fishing activity and the duration of the activity at each fishing site. Indicated is the family and the species of the caught fish (no catch indicates sites where no fish was caught). Furthermore, it shows the total length of the caught fish (TL) and whether it belonged to landings or to bycatch. The catch category indicates what happened to the fish after it was caught (catch = landed, used as bait, discarded dead, discarded alive or personal use). Finally, it shows the fitting parameters of the length-weight relationship of each species and the estimated weight of each individual that was caught during the monitoring.

Click here for additional data file.

We express our gratitude to the fishers associations, the “Unión de Cooperativas de Producción Pesquera Artesanales de Galápagos” (COPROPAG) and the “Unión de Cooperativas de Pesca de Galápagos” (UCOOPEPGAL) for their collaboration and all participating fishers who helped us with their collaboration at sea and by sharing their knowledge during the interview surveys. We are grateful to the Charles Darwin Research Station and the Galápagos National Park Services for providing the necessary logistics. This publication is contribution number 2109 of the Charles Darwin Foundation for the Galapagos Islands.

Additional Information and Declarations

Competing Interests

Author Contributions

Human Ethics

Field Study Permissions

The authors declare there are no competing interests.

Johanna S. Zimmerhackel conceived and designed the experiments, performed the experiments, analyzed the data, wrote the paper, prepared figures and/or tables, reviewed drafts of the paper.

Anna C. Schuhbauer conceived and designed the experiments, performed the experiments, contributed reagents/materials/analysis tools, prepared figures and/or tables, reviewed drafts of the paper.

Paolo Usseglio performed the experiments, analyzed the data, contributed reagents/materials/analysis tools, prepared figures and/or tables, reviewed drafts of the paper.

Lena C. Heel conceived and designed the experiments, performed the experiments, prepared figures and/or tables, reviewed drafts of the paper.

Pelayo Salinas-de-León analyzed the data, contributed reagents/materials/analysis tools, prepared figures and/or tables, reviewed drafts of the paper.

The following information was supplied relating to ethical approvals (i.e., approving body and any reference numbers):

The research was approved by the Galapagos National Park under the annual research plan of the Charles Darwin Foundation (POA 2012, number 86).

The following information was supplied relating to field study approvals (i.e., approving body and any reference numbers):

The research was approved by the Galapagos National Park under the annual research plan of the Charles Darwin Foundation (POA 2012, number 86).

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
