# Peer review of "Catch, bycatch and discards of the Galapagos Marine Reserve small-scale handline fishery"

_PeerJ, doi:10.7717/peerj.995_

## Round 0.1 · original submission · Minor Revisions

Both reviewers have been very positive about this paper and I agree with them. It seems quite incongruous that such an iconic area has such limited fisheries management measures. I have annotated a version of the pdf. Many comments are just minor grammatical corrections and some are suggestions that you may not agree with.

·

Basic reporting

The reporting was generally fine. However there were a few sentences that needed clarification (see pdf and word doc for comments), also below...

Abstract – See suggestions, wasn’t clear whether the bycatch ratio was by weight or numbers, I know it’s implicitly the former though perhaps better to make explicit. Your percentages were not clear in terms of what they referred to (i.e. number of species of shark classified as bycatch, total numbers, or weight?). Perhaps better to say “target catch” than “catch composition”.

For recommended management measures wasn’t too sure whether all of these flowed from your research findings. E.g. you recommend spatio-temporal closures, though your study doesn’t really have a spatial or temporal component.

Introduction

Line 34: I’m unsure why bycatch is especially an issue where serious ecosystem degradation has already been observed? Though you clarify this in line 41.
Line 64: Substitute “artisan” for “low technologically advanced”.
Line 65: Not too sure if I agree with this statement. Many are often unregulated and not managed, this a particular problem in small-scale fisheries where there is multiple access.

Experimental design

Appears technically competent, though see below:

Line 167: For your prediction of bycatch sizes I wondered if you could take into account the length of maturity for the species you studied (a crude estimate could probably be found on Fish Base)? This would be particularly informative for management as it would allow you to identify which species are being caught before they have chance to reproduce.

Validity of the findings

Findings appear to be valid.

Additional comments

Generally think that there needs to be a more thorough discussion of what the data you collected means for specific management measures, and less drawing on case-studies elsewhere. You could maybe suggest what further research would be needed to improve management?

·

Basic reporting

This is a fairly straightforward and solid piece of work that was relatively easy to interpret. As the first study of it's kind on this fishery (somewhat surprisingly) this work will certainly serve as a very useful reference point for consideration of how to improve management of the fishery. The text was relatively well written and appropriately referenced. Findings were also clearly presented and soundly interpreted. As such I believe this is a worthwhile study which should be published. However I do have some specific comments on the text along with suggestions with how it should be improved in certain places.

Introduction:
Good overview. No mention of any highly protected (i.e. no-take) areas. Do any exist? If so, how do they influence the fishery? (see also comments on the discussion section).

Results:
It would be good to have a measure of variability (either SD or SE or range of estimates) for some of the results which have been reported. For example, some fishers may have used most of the catch for personal consumption while other may have sold it all.
In terms of numbers of individuals discarded (or used for bait) a figure of 77 % is very high! This result should definitely feature in the discussion (see below).
Mycteroperca olfax – most commonly retained species – considered vulnerable on the IUCN Redlist.
Figure 3 – it would be interesting to see how the size at 80% probability of retention compares to size at maturity (e.g. 50% mature) for each of these key species. Ideally this could be put on the graphs as a line or similar.

Discussion:
The paper is about both catch and bycatch – so don’t focus quite so much just on the bycatch angle (see below, comment on line 329).
It is worth reporting bycatch ratios in terms of both biomass and numbers (see above). Numbers of individuals discarded (or used for bait) are potentially more biologically and economically meaningful, especially for targeted species.
Line 302 – missing the “d” off “discard”.
Line 316 – you might like to add the reference: Diamond B & Beukers-Stewart BD (2011) Fisheries discards in the North Sea: Waste of resources or a necessary evil? Reviews in Fisheries Science. 19(3): 231-245
Line 329 – Species composition: I would actually recommend starting the discussion with this section.
Line 355 – insert a space after Lewison et al 2004)

Experimental design

Monitoring was only conducted between February and May 2012 – how might catches vary at other times of the year or even between years? This needs to be covered in the discussion.
What proportion of the total number of fishers and / or fishing trips were monitored? How were they selected? Detail on these questions should be reported here.
Excellent effort to collect all possibly relevant information
Nice categorisation of catch / bycatch into different components
Likewise, what proportion of fishers were interviewed? OK, I now see you reported this in the results section, but it might be better here in methods.
Specific details (wording) of questions / questionnaire used are missing. These should be provided.
Likewise we need a bit more details about how the coding was done.

Validity of the findings

The general findings of the study all appear fairly well justified by the data – but some issues with experimental design need to be incorporated into the discussion (see above).
Management suggestions were carefully thought through and all appear sound.
However, you don’t mention using no-take marine reserves to protect essential habitat and or spawning refuges / aggregation sites. Do any of these currently exist and would it be beneficial to increase their use in the future?

Additional comments

No comments

---

## Round 0.2 · accepted · Accept

Sorry about the delay - I have been travelling and had variable internet connectivity. You have answered and dealt with the queries and suggestions very well and I think that the referees comments have had a positive impact on this paper.